# Effect of Different Protection on Lateral Ankle during Landing: An Instantaneous Impact Analysis

**DOI:** 10.3390/bioengineering10010034

**Published:** 2022-12-27

**Authors:** Junchao Guo, Jiemeng Yang, Yawei Wang, Zhongjun Mo, Jingyu Pu, Yubo Fan

**Affiliations:** 1Beijing Key Laboratory of Rehabilitation Technical Aids for Old-Age Disability, Key Laboratory of Human Motion Analysis and Rehabilitation Technology of the Ministry of Civil Affairs, National Research Center for Rehabilitation Technical Aids, Beijing 100176, China; 2Key Laboratory for Biomechanics and Mechanobiology of Ministry of Education, School of Biological Science and Medical Engineering, Beihang University, Beijing 100191, China; 3The Second Center Kindergarten of Yinghai Town, Daxing District, Beijing 100176, China

**Keywords:** biomechanical behavior, ankle inversion, dynamic stability, ankle brace, finite element

## Abstract

Ankle sprain is the most common injury during parachute landing. The biomechanical behavior of the tissues can help us understand the injury mechanism of ankle inversion. To accurately describe the injury mechanism of tissues and assess the effect of ankle protection, a stable time of landing was obtained through the dynamic stability test. It was used for the boundary condition of the foot finite element (FE). The FE model was provided a static load equal to half of the bodyweight applied at the distal tibia and fibula; a foot-boot-brace FE model was established to simulate the landing of subjects on an inversion inclined platform of 0–20°, including non-, external, and elastic ankle braces. Compared with the non-ankle brace, both the external and elastic ankle braces decreased the peak strains of the cal-fibular, anterior Ta-fibular, and posterior Ta-fibular ligaments (15.2–33.0%), and of the peak stress of the fibula (15.2–24.5%). For the strain decrement of the aforementioned ligaments, the elastic brace performed better than the external ankle brace under the inversion of the 10° condition. The peak stress of the fibula (15.6 MPa) decreased up to 24.5% with an elastic brace and 5.6–10.3% with an external brace. The findings suggested that the behaviors of lateral ankle ligaments and fibula were meaningful for the functional ability of the ankle. This provides some suggestions regarding the optimal design of ankle protection.

## 1. Introduction

Ankle injuries are the most common foot injuries during paratrooper landings [1], with 74% of all ankle injuries being ligamentous sprains [2]. Ankle ligament injuries are also reported annually in US army paratroopers. They account for 30–60% of parachuting-related injuries in the military [3]. Moreover, 85% of ligament sprains are caused by an inversion trauma while performing exercises or during the landing phase [4]. The cost for treatment and rehabilitation per sprain is also higher, and the dollar costs ranged from $318 to $941 per patient [5]. However, biomechanical mechanisms underlying ligament injuries with a sudden ankle inversion have not been well addressed in landing on an inclined plane [3].

Half-squat parachute landing in the Chinese Air Force is a kind of activity with high impact force [6]. The abnormal impact can change the kinematics and plantar loading of the lower extremities and increase injury risk [7]. Factors causing ankle ligament injuries in foot inversion have been reported using many methods in exercises or training. The risk factors are classified as both intrinsic and extrinsic [8]. Extrinsic factors include postural alignment, training error, equipment, age, height, body weight, sex, and environmental conditions. For fatigued recruits or sportsmen, the extrinsic condition is associated with a high risk of ankle-inversion sprains. In particular, injured taller and heavier recruits have higher morbidity of lateral ankle sprain based on their training or on actual combat [9]. Moreover, the different dropping heights and the inversion velocity exhibited some correlation with each other during parachute landing [10,11]. The intrinsic factors could involve physical characteristics such as muscle strength imbalance, the limited range of joint motion, joint instability, and generalized joint laxity. From the perspective of ankle-joint anatomy, Hicks showed that the subtalar joint contributed to the foot rotation around three axes. The oblique axis from the subtalar joint caused the foot to pronate and supinate [12]. This functional instability of the ankle joint leads to a loss of self-control of the motion axis [13]. It also induces muscle strength imbalance, thereby increasing the incidence of ankle-inversion sprains [12,13]. The imbalanced muscle force was found to be a risk factor in military training [9,14]. However, the ligaments injuries are only diagnosed on the basis of statistics and imageology. This quantified analysis of ligamentous injury is not well understood in terms of these risk factors [15].

To decrease the effect of intrinsic and extrinsic factors on ankle inversion injury, the different ankle braces were developed in army or sport. A hard plastic external brace with an air cushion in medial and lateral ankle locations was first used in 1993 to prevent excessive ankle inversion. The results suggested that the ankle brace could effectively reduce the incidence of ankle sprains in parachute landing by 85% [16]. Moreover, it also became clear that restricting the inversion angle was better for stabilizing the balance of the ankle [17]. After using a semi-rigid brace, the subtalar joint exhibited a tendency of normal ankle kinematics [18]. The semi-rigid brace reduced ankle and subtalar joint motion of dorsiflexion and kept the hindfoot in a sagittal motion [19], while lace-ups and stirrups appear to be more effective at preventing and treating lateral ankle sprains [20]. The brace also reduced talar tilt and passive motion in the coronal plane [21]. Despite the subtalar joint kinematics, the ankle and hindfoot were investigated, with the cadaveric foot placed in neutral, dorsiflexion, and plantarflexion positions [4,9,13]. However, the dynamic behavior of the lateral ligament sprain during parachute landing was not well detected or investigated.

There are more risk components to the sprained ankle inversion than the factors of other types of ankle injury. To further avoid the risks involved in parachute training, the different protection of ankle braces was applied in the initial military affairs. For example, the statistical methods were used to compare the effect of different ankle braces in preventing ankle inversion during a basketball rebounding task [14], including the novel prophylactic ankle brace [22]. The mechanism of injury is unclear, due to the multicomponent involvement of the ligamentous injury, sprained ligaments, articular contusion, and a complex landing platform, all of which are not comprehensively represented in the current literature. To assess the biomechanical mechanism underlying lateral ankle injury and improve the reliability of the ankle brace, this study (1) investigated the mechanism of ankle lateral ligament injury and (2) described the protective effect of different ankle braces (external and elastic braces) during parachute landing on an inclined platform of 0–20°.

## 2. Materials and Methods

### 2.1. Subjects and Experimental Measurement

A total of 16 adult male paratroopers were enrolled as study participants after they signed the informed consent with Beihang University. These volunteers (average: 71.3 ± 5.4 kg and 25 ± 1.8 years old) had no acute lower extremity injury or previous ankle injury or surgery. Subjects with ankle braces, boots, and no braces were made to stand on the predefined platform and jumped down from different height platforms (152 cm, 82 cm, and 52 cm) on a force plate [6,10] (FP4060-08-100, Bertec, Inc., Columbus, OH, USA) (Figure 1). One foot landed on the board, and the other foot was on the force plate. The data of the ground-reaction force was recorded with a 1000 Hz sampling frequency of the force plate. To establish the external ankle brace and boot model, their point cloud date was obtained using 3D Scanner equipment (Tianyuan, Inc., Tianjin, China). An F-scan system measured the plantar pressure of subjects standing on a 20° inclined platform. The partitions of plantar pressure in F-scan software were divided into 10 regions: the medial and lateral forefeet, and the first to fifth metatarsal, midfoot, and medial and lateral hindfeet [23].

### 2.2. Data Acquisition and Analysis

Subjects jumped down from the different height platforms on the force plate. The stable time of subjects landing for each test was evaluated using the sequential estimation method. The balanced condition during foot landing was observed when the standard deviation of the overall average was *p* < 0.05 [24]. Their landing time had a tendency towards consistency due to the professional nature of the subjects. Each subject performed three jumps from the different platform heights. Means and standard deviations were used for the analysis of dynamic stability. The time of dynamic stability [25] was approximately 87–95 ms (Figure 2). This value is used for the boundary condition of foot finite element (FE).

### 2.3. FE Model

Geometric outline of the left foot was obtained through the three-dimensional reconstruction of magnetic resonance images in the unloaded position (71 kg). Then, the images were imported and segmented to acquire the boundary of each bone and skin surface in MIMICS10.01 (Materialise, Leuven, Belgium). The geometries of skeleton and skin were processed using RapidForm software (RapidForm INUS Corporation, Seoul, Republic of Korea) to form solid structures of each component. The established foot model included 28 bones (the distal of the fibula and tibia), articular cartilages, major ligaments, Achilles’s tendon, deep fascia and superficial fascia, and skin (Figure 3a). The point cloud of the boot profile was processed in Geomagic12.0 (Figure 3b) (Geomagic Studio, Springfield, MA, USA). The elastic ankle brace was obtained based on the surface of the skin profile (Figure 3c). The external ankle brace from the 3D scanner equipment data was processed and is shown in Figure 3d. The solid model of each component was imported and assembled in the FE package (ABAQUS 6.13, SIMULIA Inc., Providence, Johnston, RI, USA). The constraint relationship among the internal bones, which manifested as contact action, was applied to the interaction of components, which allowed for relatively accurate movement. The contact option of surface-to-surface in software was used in the relationship of joint setting. To simulate the covering layers of articular cartilages, compressive properties resembling cartilaginous material was used between each pair of joint contact surfaces [23,26]. The interaction between the ankle brace and the skin was defined by the pretension force in ABAQUS. All the bony, heel-fat-pad, ligamentous, and cartilaginous components were completely embedded in a volume of soft tissues in the skin.

The material mechanical properties of the bone tissue [27], deep fascia and superficial fascia [28], cartilages [29], and ligaments [30] were determined from the literature. The material properties of the ankle brace and shoe sole were defined with polyvinyl chloride and ethylene-vinyl acetate copo to enable the use of impact resistance, respectively (Table 1). The encapsulated soft tissue [31] was defined as a hyperelastic material. A second-order polynomial strain energy potential (ABAQUS 6.11, SIMULIA Inc., Providence, Johnston, RI, USA) was adopted with the following Equation:(1)$$U=\sum_{i+j=1}^{2}C_{ij}{\left({\bar{I}}_{1}-3\right)}^{i}{\left({\bar{I}}_{2}-3\right)}^{j}+\sum_{i-1}^{2}\frac{1}{D_{i}}\left(J_{e1}-1\right){\left(J_{e1}-1\right)}^{2i}$$
where U is the strain energy of the reference volume per unit; $C_{ij}$ and $D_{i}$ are corresponding material parameters of the constitutive equations (Table 2); and ${\bar{Ι}}_{1}$ and ${\bar{Ι}}_{2}$ are the first and second deviatoric strain invariants defined as follows:(2)$${\bar{Ι}}_{1}={\bar{\lambda_{1}}}^{2}+{\bar{\lambda_{2}}}^{2}+{\bar{\lambda_{3}}}^{2}\;$$
(3)$${\bar{Ι}}_{2}={\bar{\lambda_{1}}}^{-2}+{\bar{\lambda_{2}}}^{-2}+{\bar{\lambda_{3}}}^{-2}\;$$
with the deviatoric stretches $\bar{\lambda_{i}}$=Je1−13λi.  Je1 and $\lambda_{i}$ are the elastic volume ratio and the principal stretches, respectively. 

The very rigid contact interface of the 0–20° inclined plate was used to simulate the landing of subjects. The contact interaction between the foot and support was established with minimal induced pressure before the applied loading conditions. Approximately half of the body weight of the subject (31.5 kg) was loaded onto the distal of the tibia and the fibula. The stable time value (Figure 2) as a boundary condition was used to simulate the impact time between the foot and the platform.

## 3. Results

The FE model was validated by the results of F-scan measurements taken while standing on a 20° inclined platform. The peak stress of 9.7 MPa from the prediction concentrated on five metatarsal regions, whereas the value in the F-scan measurement increased to 10.12 MPa, and the pressure regions of peak stress concentrated on the lateral metatarsal region of forefoot (Figure 4). The measured peak pressure was greater than the value predicted through the simulated calculation. This difference may be due to the resolution ratio of the F-scan sensors, which reported an average pressure for an area of approximately 25 mm^2^.

Figure 5 showed peak stresses of lateral ankle ligaments (the calcaneofibular ligament-CFL, the anterior talofibular ligament-ATFL, and the posterior talofibular ligament-PTFL) under different ankle-brace conditions when the foot landed on an inclined platform of 0–20°. Compared with the non-ankle-brace protection, the external-ankle-brace protection decreased the peak stresses of CFL, ATFL, and PTFL by 25.4–32.3% (Figure 5a), 15.2–33.0% (Figure 5b), and 22.4–27.2% (Figure 5c), respectively. The predicted peak stresses of the aforementioned ligaments with the external ankle brace were lower than those with the elastic ankle brace (Figure 5). The peak stress of CFL had an obvious increase (Figure 5a). The increase was approximately 72.4–83.6%. The peak strains of the aforementioned ligaments are shown in Figure 6. Compared with the non-ankle-brace condition, the peak strains of CFL, ATFL, and PTFL decreased by 4.1–21% (Figure 6a), 27.0–37.3% (Figure 6b), and 14.3–53.9% (Figure 6c), respectively, under the external ankle-brace condition, and by 3.5–24.3% (Figure 6a), 14.7–24.1% (Figure 6b), and 18.2–51.0% (Figure 6c), respectively, under the elastic-ankle-brace conditions.

The stress distribution and peak stresses of the fibula under the non-, elastic-, and external-ankle-brace conditions are shown in Figure 7 when the foot landed on an inclined platform of 0–20°. The same stress-distribution tendency of the fibula concentrated on the distal of the fibula and on approximately a third of the proximal fibular neck (Figure 7a). Compared with the non-ankle-brace condition, the peak stress of the fibula (15.6 MPa) decreased by approximately 24.5% in the 20° incline under the elastic-ankle-brace conditions. The peak stress of the fibula under the external-ankle-brace conditions decreased by 10.3% in the 5° incline and 5.6% in the 20° incline (Figure 7b).

## 4. Discussion

In this study, the computational model of the foot was validated by the F-scan measurement results (Figure 4). The model was mainly used to evaluate the biomechanical behaviors of the lateral ankle ligaments and the fibula during paratrooper landing on a varus incline of 0–20°. The pressure regions from FE prediction and F-scan measurements were similar while standing on a 20° inclined platform, but the magnitudes of the values were not completely equal. The plantar pressure value of the forefoot in experimental testing was slightly greater than the FE-prediction value. This discrepancy may have been due to the difference in the measurement principle and FE-analysis theory. Moreover, the simulation calculation ignored the interaction between the internal bones and surrounding soft tissues.

Previous studies have indicated that the ankle brace is a cost-effective intervention for reducing the incidence of ankle injuries, sprains, and fractures during paratrooper landing [32]. In particular, the external ankle brace consisting of a plastic external shell with air bladders prevents extreme ankle inversion/eversion. In a survey of the US army, this brace was found to decrease ankle sprains by 9–33% [33] and lower the risks of ankle injury by 40% [34]. This result was consistent with the predicted results that the ankle-brace protection was greater than non-ankle-brace protection during paratrooper landing on the incline of 0–20° (Figure 7). The maximum decrement in the peak stress of the fibula was 24.5% with the elastic ankle brace compared with the non-ankle brace while landing on the 20° incline. The elastic brace effectively decreased the effect of the external load on the fibula to avoid further injury. Thus, the risk of fibular injury was reduced to a certain extent [33]. The risk of corresponding CFL, ATFL, and PTFL fractures decreased because of the anatomical attachment point on the fibula [35]. The elastic brace might also stabilize the ankle balance in chronic ankle instability patients [36]. However, the injury mechanism of internal ligaments and bone tissue could not be completely determined through in vitro testing by using elastic and external ankle braces.

In fact, lateral ankle ligament sprains are the most common injuries during paratrooper landing on a varus incline [32]. Researchers have shown a functional effect of the ankle brace on the lateral ligament [37,38]. In this study, the FE-predicted results showed that the peak strain decrement of ATFL (37.3%) was greater with the external-ankle-brace protection (24.1%) than with the elastic-ankle-brace protection (Figure 6b). The external ankle brace effectively protected the excessive tensile of ATFL [37]. This result was consistent with that of the study wherein the external ankle brace reduced the injury rate of 50% ankle sprains compared with the non-ankle brace [16]. The brace re-established ankle stability from the plantar flexion to internal rotation in sudden inversion [34,35]. However, the strain decrement of CFL with the elastic-ankle-brace protection (24.3%) was greater than that with the external-ankle-brace protection (21%) on a 7–13° incline (Figure 6a). This may have been due to the physiologic function of the ankle in different ankle postures [39]. The external and the elastic ankle braces prevented the excessive tensile of PTFL (Figure 6c). This result was consistent with the previous study that the PTFL could not be torn with the ankle inversion if not for extreme dorsiflexion [34,39].

Despite the fact that the ankle brace reduced the sprain of the ankle ligament [32], this study showed the limited protection of the ankle brace with the increase in the incline angle. From Figure 5 and Figure 6, both strains and stresses of CFL, ATFL, and PTFL had a tendency to increase gradually. This supported the notion that the ATFL and CFL function together at all positions of ankle flexion to provide the stability of the lateral ankle [39]. Of course, the muscle tuning was also important to maintain the stability of soft tissue (ankle ligaments) [40,41]. For the stress and strain of PTFL, there was a tendency to decrease, especially in the protection of the external ankle brace [34]. This was related to ATFL strain increasing in the external rotation of ankle joint [39]. It was also shown that the lower extremity muscle activity was adjusted in reaction to impact force with the goal of minimizing soft-tissue vibrations [42]. Therefore, the paratrooper landing impact would benefit from the ankle brace [34], the landing posture [10], and the lower extremity muscle tuning [40,41,42].

From the FE-predicted results in Figure 7, the stress distribution of the fibula was concentrated on the distal location and approximately a third of the proximal fibular neck under the three ankle-brace protections (Figure 7a). The previous study showed that approximately 11% of fractures occurred in the ankle joint during paratrooper landing. The external ankle brace reduced the fracture rate to 4–5% [3]. This indicated that brace protection improves ankle stability in different ankle postures [39] and decreases the risk of fibular fracture. The FE-predicted results also showed that the peak stress of the fibula (15.6 MPa) decreased by 24.5% with the elastic ankle brace and by 5.6% with the external ankle brace (Figure 7b). The fibula as the anatomical attachment point of CFL, ATFL, and PTFL ligaments was stretched in the plantar flexion or ankle rotation motion during sudden inversion [35]. Figure 5 showed that the lower peak stresses of ligaments under the external and elastic-ankle-brace conditions decreased against the tensile ability of the fibula compared with those under the non-ankle-brace conditions. The brace maintained the ankle stability to a certain extent [39]. The lower peak stress of the fibula did not lead to neck or distal fracture [33]. Both ankle braces maintained the joint stability of ankle inversion [36], thereby reducing the risk of fibular fracture [35].

As an FE simulation method based on simplifications, the present study had some limitations. First, the material properties of tissues in the FE model might be different from the real tissue structure to some extent, but information in the impact test condition would need to be verified in the future. Second, the effect of the friction between the ligaments and surrounding soft tissues on the contact relationship was unclear in the present literature. Third, only the Achilles tendon, except for the distal fibula and tibia, was loaded by half of the one foot bearing weight. The reasonableness of this value for a impact test was debatable.

## 5. Conclusions

In this study, the biomechanical behaviors of lateral ankle ligaments and the fibula under the protection of external and elastic ankle braces were quantified in comparison with a non-ankle brace. The results of this study indicate: (1) Compared with paratroopers landing on an inclined platform of 0–20° under the non-ankle-brace condition, the peak stress of the lateral ankle ligaments and the fibula decreased in those landing on an inclined platform of 0–20° under the external and the elastic-ankle-brace conditions; (2) The external ankle brace was more effective at reducing the peak stress and strains of ATFL and PTFL, especially regarding the peak-strain decrease in PTFL during the foot landing on 10–20° incline; (3) The external-ankle-brace protection against the peak stress of the fibula was lower than the elastic brace protection. These findings indicated that the combination of external and elastic ankle braces might constitute more effective protection for paratrooper landing.

## Figures and Tables

**Figure 1 bioengineering-10-00034-f001:**
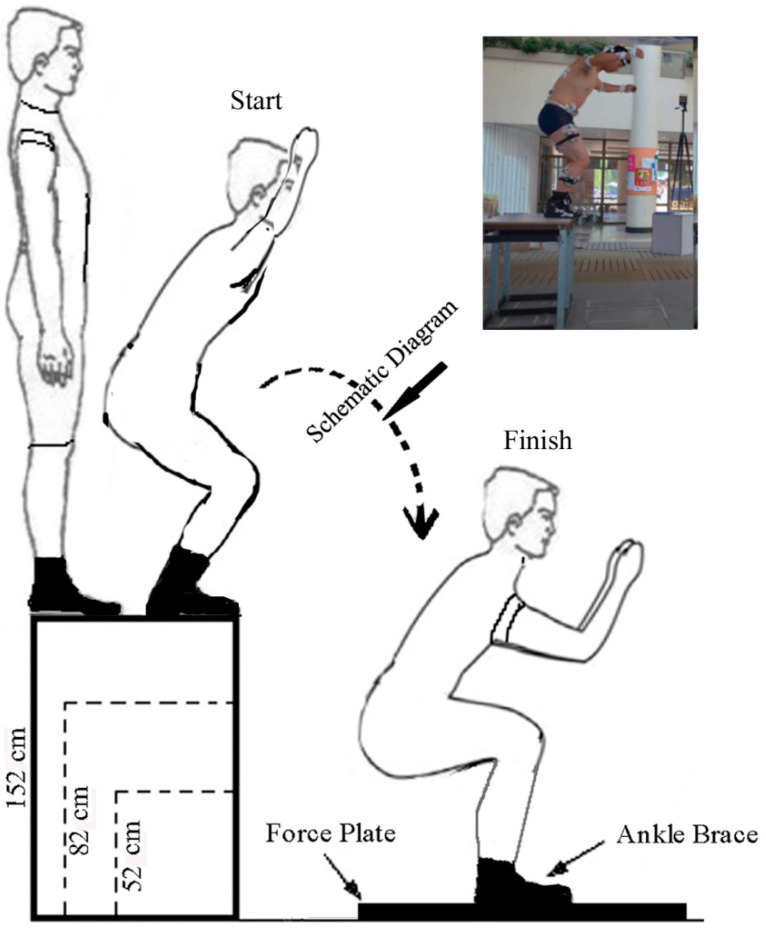
Landing on force plate from the different height of platform (subjects with ankle braces, boots, and no braces).

**Figure 2 bioengineering-10-00034-f002:**
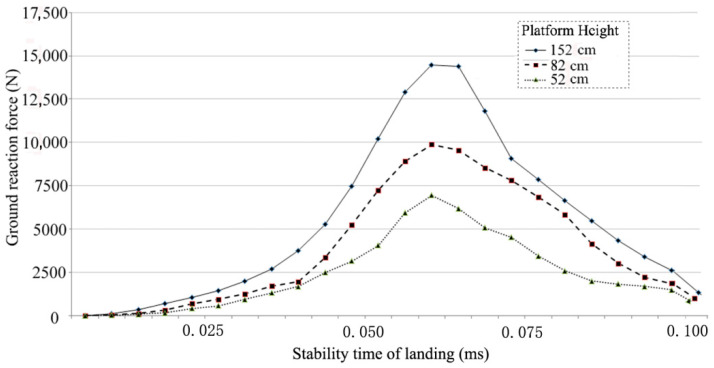
The dynamic stability time of subjects from the different height platforms (52, 82, and 152 cm).

**Figure 3 bioengineering-10-00034-f003:**
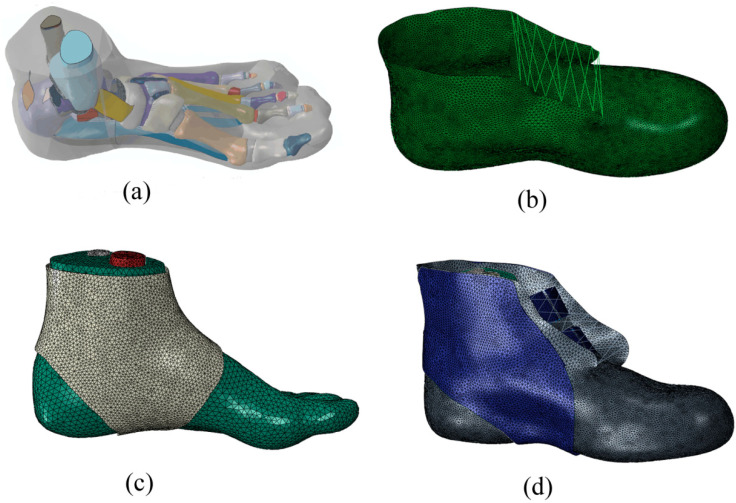
FE model: (**a**) foot structure; (**b**) boot; (**c**) foot and the elastic ankle brace; and (**d**) foot, boot and the outside ankle brace.

**Figure 4 bioengineering-10-00034-f004:**
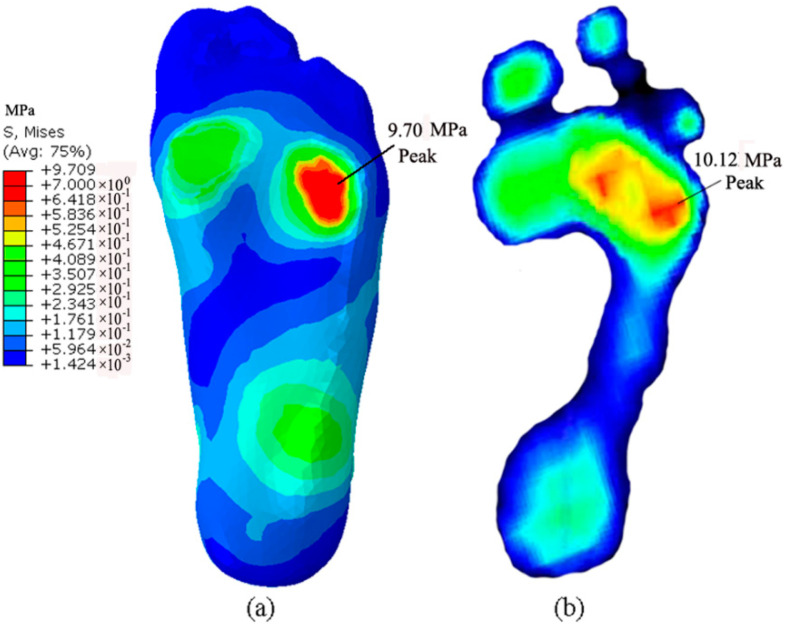
The plantar pressure standing on 20° incline: (**a**) FE model prediction; (**b**) F-scan measurement.

**Figure 5 bioengineering-10-00034-f005:**
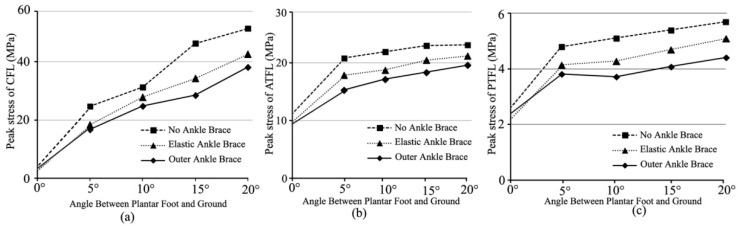
Peak stresses of the lateral ankle ligaments during the foot landing on 0–20° incline: (**a**) CFL; (**b**) ATFL; and (**c**) PTFL under non-, elastic-, and external- ankle-brace conditions.

**Figure 6 bioengineering-10-00034-f006:**
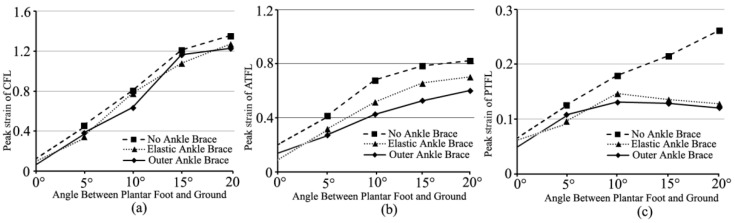
Peak strains of the lateral ankle ligaments during the foot landing on 0–20° incline: (**a**) CFL; (**b**) ATFL; and (**c**) PTFL under non-, elastic-, and external-ankle-brace conditions.

**Figure 7 bioengineering-10-00034-f007:**
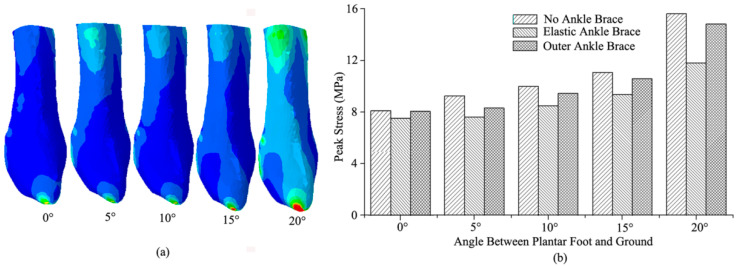
Peak stress and stress distribution of the fibula when the foot landing on 0–20° incline: (**a**) the stress-distribution tendency; (**b**) peak stress under non-, elastic-, and external-ankle-brace conditions.

**Table 1 bioengineering-10-00034-t001:** Material mechanical properties and element types of components in FE model.

Component	Element Type	Young’s Modulus (MPa)	Poisson’s Ratio	Cross-Sectional Area (mm^2^)
Bony structures	3D-tetrahedra	7300	0.3	—
Soft tissue	3D-tetrahedra	Hyperelastic	—	—
Cartilage	3D-tetrahedra	10	0.4	—
Ankle ligaments	3D-hexahedron	0~700	—	15~70
Other ligaments	Tension-only truss	260~350	0.49	28~170
Fascia	3D-hexahedron	350	0.3	290.7
Shoe sole	3D-tetrahedra	913	0.37	—
Ankle brace	3D-tetrahedra	3150	—	—
Plantar support	3D-hexahedron	25,000	0.1	—

**Table 2 bioengineering-10-00034-t002:** The coefficient of the hyperelastic material model used for the encapsulated soft tissue.

$C_{10}\;$	$C_{01}\;$	$C_{20}\;$	$C_{11}\;$	$C_{02}\;$	$D_{1}\;$	$D_{2}\;$
0.08556	−0.05841	0.03900	−0.02319	0.00851	3.65273	0.0000

## Data Availability

All data included in this study are available upon request by contact with the corresponding author and the first author.

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
