# Peer review of "Effect of Different Protection on Lateral Ankle during Landing: An Instantaneous Impact Analysis"

_bioengineering, 2022, doi:10.3390/bioengineering10010034_

Round 1

Reviewer 1 Report (Previous Reviewer 1)

good job

Author Response

Response to Reviewer 1 Comments

Dear Reviewer:

We will revise the grammatical error and language style in whole manuscript. Thank you very much for your suggestion and patience again.

May happiness follow you wherever you go!

Kind regards,

Dr Guo

Reviewer 2 Report (Previous Reviewer 2)

I don' t have comments for the Authors

Author Response

Response to Reviewer 2 Comments

Dear Reviewer:

Thank you very much for your suggestion and patience again.

May happiness follow you wherever you go!

Kind regards,

Dr Guo

Reviewer 3 Report (Previous Reviewer 3)

General Comments:

The goal of this investigation was to examine ankle ligament loading when landing from a jump onto an inclined surface, simulating an inversion ankle sprain.  Specifically, they are interested in paratroopers landing.  They collected some ground reaction force data of paratroopers landing from different heights, all on a flat surface.  It appears the only thing the landing data was used for was to obtain time to stability.  It appears that the model was only provided a static vertical ground reaction force equal to half a subject’s bodyweight through the tibia and fibular (though there is reference to providing it through the Achilles at the end of the Discussion).  If this is not the case, it needs to be clarified in the Methods as to what dynamic ground reaction force profile was provided as input to the model. It also appears that the model assumed flat-footed contact with the ground.  When landing a person does not land flat footed, they first land on their forefoot and eventually stand flat-footed at the end. 

Overall, I’m still very concerned how their model accounted for (or did not account for) the highly dynamic nature of landing from a jump.

The authors provided a relatively detailed response to my concerns.  However, many of them did not get incorporated into the manuscript.  I don’t want responses just to please me, they need to be incorporated into the manuscript.  I also don’t think that the athors understood some of my questions.  Some additional specific comments follow.

Specific Comments:

Abstract

11)      Line 15, it is clear that you are interested in paratroopers.  Why not lead with that in your abstract?  Your first sentence is about exercise or sport, which does not include landings of a paratrooper.

22)      Line 17-18, it should be indicated that your FE model was provided a static load equal to half a person’s bodyweight applied at the distal tibia and fibula. 

33)      Line 23 and 26, ‘fibular’ should have ‘ligament’ after it.

Introduction

11)      Line 34, ‘They’ not ‘It’.

22)      Line 34, ‘The cost for treatment and rehabilitation per sprain is also higher.’  Still not clear, higher than what?  Maybe you should just say ‘high’ since you aren’t comparing to anything.

33)      Line 46-47, which or the extrinsic conditions are you referring to when fatigued? Lines 47-48, you indicate height and body weight, but neither of these are in your list of extrinsic factors.

44)      Line 55, is pronation and supination different from eversion and inversion?  You indicated in your response to my original review that it is different, but the image you provided did not have pronation and supination on it, only eversion and inversion.  If they are different, make sure your text indicates how they are related in eversion and inversion, since inversion is your motion of interest related to ankle sprains.

55)      Line 77-89, you indicated in your response to my previous review that you added hypotheses and made the Introduction hypothesis driven (i.e., you funnel the reader to the logical conclusion that your work is needed and the hypothesis is supported by the literature).  There are still no hypotheses!

Materials and Methods

11)      Line 95, should add that boots were worn.

22)      Line 96, would still be helpful to indicate the instructions provided to the subjects on how to jump and land (this could be added to the Figure 1 caption). 

33)      Line 113, Were the three jumps averaged to create a representative average for each person?  How was this data then combined to create the dynamic vertical ground reaction force profile that served as input to the FE model? 

44)      Line 98, did you filter the ground reaction forces or were they raw data?  You did not answer this in your response to my original review and it does not appear to be indicated in the manuscript if the data was filtered.

55)      Line 99-100, when was the scanner used, was it just when they were statically standing (on an inclined platform or level ground), or was it able to measure them dynamically during their drop landings?

66)      Line 111, what are the units on 0.25? Not sure what you mean by ‘Quantitative data would firstly be shown by Mean value.’

77)      Line 113, Were the three jumps averaged to create a representative average for each person?  How was this data then combined to create the dynamic vertical ground reaction force profile that served as input to the FE model? 

88)      Line 118-119, I’m still confused by how you refer to the horizontal axis as ‘stability time’.  It is just ‘time’ with stability occurring ~0.1 seconds after initial impact.  You should also be clear, it is just vertical ground reaction force. 

99)      Did you use the horizontal forces at all in the load provided to the model?

110)   Line 121, was it just a single person that was selected for the construction of the model with a mass of 71 kg?

111)   Line 164-165, I understand that you assumed symmetry with half the landing load on each foot.  I’m more concerned by why you used a load of half body weight?  It would only be half body weight at the very end when they stabilized. Mid landing phase, as indicated in Figure 2, the vertical ground reaction force is much higher than half bodyweight on each foot.  This is what I want clarification on.  How as the dynamic nature of the landing ground reaction force incorporated into the model?

112)   I am also still interested in knowing how the location of the ground reaction force on the foot was changed during the landing and how the orientation of the foot was altered, since people don’t land flat footed!  Or, is the model of a foot flat on a surface with a static load of half bodyweight applied vertically at the ankle?

113)   I am also interested in how the horizontal forces at the ground were handled.  If you’re on an inclined surface, not all the ground reaction force will be purely vertical.  Even when landing on a flat surface there will be horizontal ground reaction forces present to keep the feet from sliding.  Is the foot attached to the ground so that it can’t slide?

Results

1)      Line 186, ‘sudden’ implies time.  There is not time component to the graphs in Figure 5a.

Discussion

1)      Without a better understanding of how the FE model utilized the dynamic conditions of landing on an inclined surface it is hard to comment on the discussion.  Landing is very dynamic, but it appears that it was treated as a highly static movement.  For example, on line 287 it is again indicated that the load was half bodyweight.  There is no debate that the forces are much greater than half bodyweight (see Figure 2)!

Author Response

Response to Reviewer 3 Comments

Dear Reviewer:

Thanks for the editors and reviewers suggestion again. We will make point-by-point responses to your comments. In addition, the revised content will be shown in Blue font corresponding to the revised manuscript based on the Red font text.

We will revise the English language and style of manuscript.

General Comments:

Point 1: The goal of this investigation was to examine ankle ligament loading when landing from a jump onto an inclined surface, simulating an inversion ankle sprain.  Specifically, they are interested in paratroopers landing.  They collected some ground reaction force data of paratroopers landing from different heights, all on a flat surface.  It appears the only thing the landing data was used for was to obtain time to stability.  It appears that the model was only provided a static vertical ground reaction force equal to half a subject’s bodyweight through the tibia and fibular (though there is reference to providing it through the Achilles at the end of the Discussion).  If this is not the case, it needs to be clarified in the Methods as to what dynamic ground reaction force profile was provided as input to the model. It also appears that the model assumed flat-footed contact with the ground.  When landing a person does not land flat footed, they first land on their forefoot and eventually stand flat-footed at the end. 

Response 1: Thanks for your comments. This comment includes all of the following questions. To make it clearer, we will make point-by-point responses to each comment. Firstly, we may not have clearly explained a fact what the landing posture of the paratroopers is in Chinese Air Force. Half-squat parachuting landing in Chinese Air Force is a prescribed and mandatory action whether training or performing tasks. The so-called Half-squat is that the feet are parallel to the ground, the upper limbs are upright, bending the knee joint and landing steadily. This posture is stipulated by our Chinese Air Force [1-2], as shown in Figure 1. In the early, both the United States and Germany adopted the forward rollover landing posture. During World War II, after Germany and the United States, the British adopt the side rollover landing posture [3-7]. Those postures are the standard action of Air Force. Therefore, paratroopers in Chinese Air Force do not firstly land on their forefoot and eventually stand flat-footed at the end. Although the rule of Half-squat parachuting landing requires the feet to be parallel to the ground, it is difficult to achieve complete parallelism. Paratroopers keep the plantar foot as parallel as possible to the ground when landing. They don't land at will like basketball players with their forefoot. Their ideology makes them to keep their plantar foot parallel to the ground. It's probably a little bit tilted to the ground. Stabilization time is the process of the plantar foot landing time on the horizontal plane. Therefore, the dynamic stability time was investigated in our manuscript. The research work is also a continuous work of our team [1, 8-12]. Including compare the landing posture and stability time of paratroopers and basketball players. Please check it.

R-Figure 1. Posture of parachuting landing [9]

  1. Niu W, Wang Y, He Y, Fan Y, Zhao Q. Biomechanical gender differences of the ankle joint during simulated half-squat parachute landing. Aviat Space Environ Med 2010, 81: 761-767.
  2. Xie X, An XZ, Tian YJ. A prospective study of 1,795 injuries of parachute landing. Journal of Preventive Medicine of Chinese People’s Liberation Army 2004, 22: 114-115 (Chinese).
  3. McNitt-Gray J. L., Yokoi T., Millward C. Landing strategies used by gymnasts on different surfaces [J]. Journal of Applied Biomechanics, 1994, 10(3): 237-252.
  4. Lord C. D., Cputts J. W. A study of typical parachute injuries occurring in two hundred and fifty thousand jumps at the parachute school [J]. The Journal of Bone & Joint Surgery, 1944, 26A(3): 547-557.
  5. Bricknell M. C. M., Craig S. C. Military parachuting injuries: a literature review [J]. Occupational Medicine (London), 1999, 49(1): 17-26.
  6. Knapik J. J., Craig S. C., Hauret K. G., et al. Risk factors for injuries during military parachuting [J]. Aviation, Space & Environmental Medicine, 2003, 74(7): 768-774.
  7. Whitting J. W., Steele J. R., Jaffrey M., et al. Does Foot Pitch at Ground Contact Affect Parachute Landing Technique? [J]. Military Medicine, 2009, 174(8): 832-837.
  8. Niu, WX; Wang, Y; Yao, J; Zhang, M; Fan, YB; Zhao, QP. Consideration of Gender Differences in Ankle Stabilizer Selection for Half-Squat Parachute Landing. Aviation, Space & Environmental Medicine. 2011, 82(12): 1118.
  9. Tianyun Jiang; Shan Tian; Xingyu Fan. Kinematics and kinetics of lower-extremity joints in parachuting landing with backpack and knee brace. Medical Engineering & Physics. 2020, 86: 1-7.
  10. Chenyu Luo, Tianyun Jiang, Shan Tian, Jie Yao, Yubo Fan. Finite element analysis of shank and ankle with different boot collar heights in parachuting landing on inversion ground surface. Computer methods in biomechanics and biomedical engineering. 2022, 25(9): 953-960.
  11. Luo, Chenyu; Jiang, Tianyun; Tian, Shan; Fan, Yubo. Finite element analysis of lumbar spine with different backpack positions in parachuting landing. Computer Methods in Biomechanics and Biomedical Engineering. 2021, 24(15): 1679-1686.
  12. Tianyun Jiang, Shan Tian, Tianhong Chen, Xingyu Fan, Jie Yao, Lizhen Wang. Protection by Ankle Brace for Lower-Extremity Joints in Half-Squat Parachuting Landing With a Backpack. Frontiers in bioengineering and biotechnology. 2021, 9: 790595.

Point 2: Overall, I’m still very concerned how their model accounted for (or did not account for) the highly dynamic nature of landing from a jump. The authors provided a relatively detailed response to my concerns.  However, many of them did not get incorporated into the manuscript.  I don’t want responses just to please me, they need to be incorporated into the manuscript.  I also don’t think that the authors understood some of my questions.  Some additional specific comments follow.

Response 2: Thanks. We'll answer questions in more detail. Not just to please you. Just like the convergence test of the finite element model. It is a necessary step in finite element analysis. However, the convergence test of finite element analysis just likes the pre-experiment. We just want to explain the more detailed work process.

Specific Comments:

Abstract

Point 3:   Line 15, it is clear that you are interested in paratroopers.  Why not lead with that in your abstract?  Your first sentence is about exercise or sport, which does not include landings of a paratrooper.

Response 3: Such is the case. We will make the point clear in the first sentence as following “Ankle sprain is the most common injury during parachuting landing”.

Point 4:   Line 17-18, it should be indicated that your FE model was provided a static load equal to half a person’s bodyweight applied at the distal tibia and fibula. 

Response 4: Yes, we will add the description in Revised Manuscript.

Point 5: Line 23 and 26, ‘fibular’ should have ‘ligament’ after it.

Response 5: Sorry, ‘fibular’ is the skeleton corresponding to Figure 7, not fibular ligament. But it is necessary to revise ‘fibular’ of the whole article.

Introduction

Point 6: Line 34, ‘They’ not ‘It’.

Response 6: Thanks, we will revise it.

Point 7:  Line 34, ‘The cost for treatment and rehabilitation per sprain is also higher.’  Still not clear, higher than what?  Maybe you should just say ‘high’ since you aren’t comparing to anything.

Response 7: Thanks for your patience. The higher of cost included five strategies [1]: (1) wrapping the ankle immediately (i.e., without obtaining further roentgenograms); (2) casting the ankle immediately; (3) obtaining stress films and providing treatment accordingly by wrapping, casting, or surgery; (4) obtaining arthrograms and performing appropriate treatment; and (5) obtaining stress films, followed by arthrograms, in patients with positive stress films and administering appropriate treatment. ‘The cost for treatment and rehabilitation per sprain’ included the above five aspects. Do we need to list everything? Please give our suggestion.

  1. Soboroff, S.H.; Pappius, E.M.; Komaroff, A.L. Benefits, risks, and costs of alternative approaches to the evaluation and treatment of severe ankle sprains. Clinical orthopaedics and related research, 1984, 183, 160-168.

Point 8:  Line 46-47, which or the extrinsic conditions are you referring to when fatigued? Lines 47-48, you indicate height and body weight, but neither of these are in your list of extrinsic factors.

Response 8: Thanks. ‘For fatigued recruits or sportsmen, the extrinsic condition would be associated with a high risk of ankle inversion sprains.’ The extrinsic conditions refers to the environmental conditions including uneven ground, potholes, raised stones, inclined contact surface, and so on. We will add the height and body weight to list of extrinsic factors. In fact, extrinsic and intrinsic risk factors included many aspects just like R-Figure 2 [1].

R-Figure 2. Extrinsic and intrinsic risk factors [1]

  1. Lysens, D.R.; Steverlynck, A.; van den Auweele, Y.; Lefevre, J.; Renson, L.; Claessens, A.; Ostyn, M. The predictability of sports injuries. Sports medicine, 1984, 1(1), 6-10.

Point 9:  Line 55, is pronation and supination different from eversion and inversion?  You indicated in your response to my original review that it is different, but the image you provided did not have pronation and supination on it, only eversion and inversion.  If they are different, make sure your text indicates how they are related in eversion and inversion, since inversion is your motion of interest related to ankle sprains.

Response 9: Thanks. They are different. R-Figure 3 showed the six basic movements of foot and ankle. Pronation is a compound movement of dorsiflexion, eversion and abduction. Supination is a compound movement of planterflexion, inversion and adduction. The landing on slope is more harmful during the parachutists landing injuries. Therefore, we analyzed and simulated the landing process. And other situations may require more research methods.

R-Figure 3. Six basic movements of foot and ankle

Point 10:  Line 77-89, you indicated in your response to my previous review that you added hypotheses and made the Introduction hypothesis driven (i.e., you funnel the reader to the logical conclusion that your work is needed and the hypothesis is supported by the literature).  There are still no hypotheses!

Response 10: Thanks. The advantage of finite element model is that the experimental results can not be sufficiently obtained. In our work, the biomechanical behavior of the internal ligament is difficult to obtain through paratrooper experiments, even dangerous to their feet. Therefore, we assess the biomechanical mechanism of the ankle inversion injury based on the many literatures of Introduction section. It should be an appropriate connection. We're not sure if it's a burden to propose hypotheses.

Materials and Methods

Point 11: Line 95, should add that boots were worn.

Response 11: Thanks. We will add the description.

Point 12: Line 96, would still be helpful to indicate the instructions provided to the subjects on how to jump and land (this could be added to the Figure 1 caption). 

Response 12: Yes, we have revised the Figure 1 to helpfully indicate the instructions of jump and land action.

Point 13:   Line 113, Were the three jumps averaged to create a representative average for each person?  How was this data then combined to create the dynamic vertical ground reaction force profile that served as input to the FE model? 

Response 13: Thanks. We found that the stability time was not more than 100ms whether pre-experiment or the formal experiment. Pressure test system (Bertec, Inc, USA) included the force plate and software. Software could record the relationship between time and force of the subject from touching force plate to the stability of the whole plantar foot. 0ms was recorded when the force plate has a value. 100ms was recorded when the force plate has a constant value. The data file of Excel format could be exported from the software. Three jumps will create a relatively reasonable average for each person. Relationship between the dynamic stability time and the vertical ground reaction force was obtained from the different height platforms (52, 82 and 152cm). It will be imported into Load module of ABAQUS analysis software (R-Figure 4). Red box in Figure 4 represents the corresponding curve of Figure 2 in manuscript.

R-Figure 4 Load module of ABAQUS software

Point 14:   Line 98, did you filter the ground reaction forces or were they raw data?  You did not answer this in your response to my original review and it does not appear to be indicated in the manuscript if the data was filtered.

Response 14: Thanks. We indeed filter the ground reaction forces. Because the high-resolution of pressure test system is up to 1000 Hz. There was a large amount of raw data. The relationship between dynamic stability time and the vertical ground reaction force has the consistent trend of curve change. Therefore, we take every 0.005 as a data interval (Figure 2). Just like the input of Figure 4 in ‘Point 13’.

Point 15:    Line 99-100, when was the scanner used, was it just when they were statically standing (on an inclined platform or level ground), or was it able to measure them dynamically during their drop landings?

Response 15: It was not able to measure them dynamical behavior. To establish the FE model, the point cloud of the ankle brace and boot was obtained using 3D Scanner equipment (Tianyuan, Inc, China) only to obtain the assembly model suitable for foot.

Point 16:    Line 111, what are the units on 0.25? Not sure what you mean by ‘Quantitative data would firstly be shown by Mean value.’

Response 16: Thank very much for your meticulous work. It was the standard deviation of overall average. The P value of standard deviation is usual 0.05 or 0.01. We used P<0.05. 0 and 2 on the keyboard are close to each other. Sorry, it was my fault. We will correct our mistake. This sentence is a little repetitive with the following sentence ‘Means and standard deviations were used for the analysis of the dynamic stability.’ Therefore, we will delete it.

Point 17:   Line 113, Were the three jumps averaged to create a representative average for each person?  How was this data then combined to create the dynamic vertical ground reaction force profile that served as input to the FE model? 

Response 17: This is the same question as ‘point 13’.

Point 18:  Line 118-119, I’m still confused by how you refer to the horizontal axis as ‘stability time’.  It is just ‘time’ with stability occurring ~0.1 seconds after initial impact.  You should also be clear, it is just vertical ground reaction force. 

Response 18: Yes, the stability time was recorded thought the pressure test system (Bertec, Inc, USA) (Software could record the relationship between time and force of the subject from touching force plate to the stability of the whole plantar foot. 0ms was recorded when the force plate has a value. 100ms was recorded when the force plate has a constant value. The data file of Excel format could be exported from the software.). Although the Chinese Air Force required the feet to be parallel to the ground when Half-squat parachuting landing, it was difficult to achieve complete parallelism. We found that the stability time was not more than 100ms whether pre-experiment or the formal experiment. In fact, parachutist landing impact injury was caused by the larger vertical force. Software recorded the time and the vertical ground reaction force, just what we obtained the stability time and the vertical force.

Point 19:  Did you use the horizontal forces at all in the load provided to the model?

Response 19: No, we did not use the horizontal force. The impact injury of parachutist landing was mainly from the vertical ground reaction force, no matter impact force on foot, knee joint.

Point 20:   Line 121, was it just a single person that was selected for the construction of the model with a mass of 71 kg?

Response 20: Thanks. This question involves research method of finite element analysis. Finite Element Analysis (FEA) simulates the real physical systems (geometry and load conditions) by mathematical approximation. It can use a finite element unit of unknowns to approximate the real system of infinite unknowns by using simple and interacting elements (i.e. elements). Therefore, FEA explains the trend of solid, liquid, fluid change. Although model of a single person was established by the FE method (no matter body weight, high, and so on), the analysis result represents the condition of paratroopers landing on the inclined plane. The values may be different based on the different body weight (60 Kg, 65 Kg, 71 Kg, 80 Kg…), but the trend is consistent. This is also the FE advantage. Response to your original review ‘Point 23’, the convergence test of the FE model and preload are work done before the comprehensive analysis.

Point 21:   Line 164-165, I understand that you assumed symmetry with half the landing load on each foot.  I’m more concerned by why you used a load of half body weight?  It would only be half body weight at the very end when they stabilized. Mid landing phase, as indicated in Figure 2, the vertical ground reaction force is much higher than half bodyweight on each foot.  This is what I want clarification on.  How as the dynamic nature of the landing ground reaction force incorporated into the model?

Response 21: Thanks. We firstly answer how as the dynamic nature of the landing ground reaction force incorporated into the model? Please check answer of ‘Point 13’. The relationship between the dynamic time and ground reaction force was imported into Load module of ABAQUS software to implement dynamic process. Secondly, we used a half of body weight based on the FE theory (just like the reviewer said the symmetry). Because paratroopers in Chinese Air Force use the posture of Half-squat parachuting landing, their feet should be parallel to the ground before landing. At the same time, the width of their feet is equal with their shoulder. We know what the reviewer confused. Figure 5 showed the board and force plate for paratroopers when landing. We embed force plate into a board, and their surfaces are parallel with no height difference. We will add the description in manuscript as following ‘One foot landed on the board, the other foot was on the force plate’.

R-Figure 5 Left foot landing on board and right foot landing on force plate

Point 22:    I am also still interested in knowing how the location of the ground reaction force on the foot was changed during the landing and how the orientation of the foot was altered, since people don’t land flat footed!  Or, is the model of a foot flat on a surface with a static load of half bodyweight applied vertically at the ankle?

Response 22: Thanks for your patience., we think reviewer should understand the landing posture stipulated of Chinese Air Force (Half-squat parachuting landing) based on the above answers ‘Point 1 and ’. The posture is different from UAS (forward rollover landing) and British (side rollover landing). Experimental test data (Figure 2 in manuscript) was used by the Load module of ABAQUS analysis software (R-Figure 4). It simulated the time frame of stability time during paratroopers landing. Load of bodyweight may be changed during the dynamic landing. However, the published literatures and our work are difficult to obtain accurate change values. The condition could be only from the published literatures [1-3]

  1. Guo, J.C.; Wang, L.Z.; Mo, Z.J.; Chen, W.; Fan, Y.B. Biomechanical behavior of valgus foot in children with cerebral palsy: A comparative study. Journal of Biomechanics, 2015, 48, 3170-3177.
  2. Jason, C.; Zhang, M.; Fan, Y.B. Three-dimensional finite element analysis of the foot during standing: a material sensitivity study. Journal of Biomechanics, 2005, 38, 1045-1054.
  3. Lemmon, D.; Shiang, T.Y.; Hashmi, A.; Ulbrecht, J.S.; Cavanagh, P.R. The effect of insoles in the rapeutic footwear: a finite element approach. Journal of Biomechanics, 1997, 30, 615-620.

Point 23:    I am also interested in how the horizontal forces at the ground were handled.  If you’re on an inclined surface, not all the ground reaction force will be purely vertical.  Even when landing on a flat surface there will be horizontal ground reaction forces present to keep the feet from sliding.  Is the foot attached to the ground so that it can’t slide?

Response 23: Thanks. We fully understand the reviewers' concerns. We did not consider the horizontal force in fact. Just like the schematic diagram of force decomposition (R-Figure 6), foot of the inclined surface is acted the ground reaction force, gravity. The horizontal ground reaction force is achieved based on the schematic diagram of force decomposition. The horizontal force is less than the impact force. Considering that the subject may be injured when landing on the inclined surface, we used the relationship between time and force (Figure 2 in manuscript) to similar to inclined surface of 0-20 °. 

R-Figure 6 Schematic diagram of force decomposition

Results

Point 24:  Line 186, ‘sudden’ implies time.  There is not time component to the graphs in Figure 5a.

Response 24: Thanks. We will revise ‘sudden’ for ‘obvious’. We want to describe the extent of increase.

Discussion

Point 25:  Without a better understanding of how the FE model utilized the dynamic conditions of landing on an inclined surface it is hard to comment on the discussion.  Landing is very dynamic, but it appears that it was treated as a highly static movement.  For example, on line 287 it is again indicated that the load was half bodyweight.  There is no debate that the forces are much greater than half bodyweight (see Figure 2)!

Response 25: Thanks. Load condition has been puzzling reviewer. Sorry! The load of FE model was from the two aspects included the half of bodyweight and the curve of stability time and ground reaction force (Figure 2 in manuscript).

Load of bodyweight may be changed during the dynamic landing. However, the published literatures and our work are difficult to obtain accurate change values. The condition could be only from the published literatures [1-3] (please refer to ‘point 22’)

The other load was the curve of stability time and ground reaction force (Figure 2 in manuscript). Please refer to ‘point 13 and 14’.

Thanks for reviewer patience again.

This manuscript is a resubmission of an earlier submission. The following is a list of the peer review reports and author responses from that submission.

Round 1

Reviewer 1 Report

1-The abstract must answer three main question, why, how, and what? So, abstract must be rewritten to contain why the authors doing this work, and what is the most important results with number and percentage and the results must be compares with the other works

2-The introduction must be containing more modern references.

3--Results paragraph need more discussions. The authors show the behavior without discussion.

4-Conclusion must be rewritten as a separated  point 

Author Response

Response to Reviewer 1 Comments

Dear Reviewer:

Thanks for the editors and reviewers suggestion. We will make point-by-point responses to yourcomments. In addition, the revised content will be shown in Red font corresponding to the revised manuscript.

Point: The paper examined the potential effect of custom made insoles on plantar pain. However justification for the study was based on assessment of metatarsal pain. Therefore study appears to have not been justified as plantar pain reduction due to insoles use has been reported previously.

Response: Thanks for reviewer comments. Many previous studies have reported the fact that the insole can reduce the plantar pain of flatfoot patients, for example the hindfoot pain, plantar fasciitis pain, the medial longitudinal arch pain, metatarsophalangeal joint pain. Our research team and I also involved with the study of plantar pressure [1-7]. We also reported the issues about plantar pressure of flatfoot patient [8]. In fact, the plantar pain of participants were mainly divided into three types: hindfoot pain, foot arch pain and forefoot pain (metatarsal regions pain). This paper wanted to focuse on investigion of the forefoot pain (metatarsal regions pain). Excuse me, is it appropriate to change the title from “plantar pain” to“forefoot pain or metatarsal pain”? Thanks for your suggestion again.

Comments and Suggestions for Authors

Point 1: 1-The abstract must answer three main question, why, how, and what? So, abstract must be rewritten to contain why the authors doing this work, and what is the most important results with number and percentage and the results must be compares with the other works

Response 1: Thanks for your suggestion. We strongly agree with you and would very much like to follow your instructions to modify. However, the editorial board of Bioengineering shows its own template and format in Figure V1. It is so difficult to know what to do.

Figure V1 Bioengineering-template

Point 2: 2-The introduction must be containing more modern references.

Response 2: Yes. We will add the more modern references and enrich Introduction section in Revised manuscript. The modern references are as following. The Introduction section was also enriched with red font.

  1. Jiang, T.Y.; Tian, S.; Chen, T.H.; Fan X.Y.; Yao, J.; Wang, L.Z. Protection by Ankle Brace for Lower-Extremity Joints in Half-Squat Parachuting Landing With a Backpack. Frontiers in bioengineering and biotechnology. 2021, 9: 790595.
  2. Dickerson, L.C.; Queen, R.M. Foot Posture and Plantar Loading With Ankle Bracing. Journal of Athletic Training (Allen Press). 2021, 56(5): 461-472.
  3. Dewar, R.A.; Arnold, G.P.; Wang, W.; Drew, T.S.; Abboud, R.J. Comparison of 3 ankle braces in reducing ankle inversion in a basketball rebounding task. 2019, 39: 129-135.
  4. Zhao, Y.J.D; Brittney, M.; Adam, J.Y.; Elizabeth, R.E.; Kang, T.H.; Kenneth J.L; Shawn F. Ankle sprain bracing solutions and future design consideration for civilian and military use. Expert Review of Medical Devices. 2022, 19(2): 113-122.
  5. Zhou, X.; Wu, D.; Wu, X.D.; Li, Z.Y.; Yan, B.; Liang, L.L.; He, Y.; Liu, Y. A novel prophylactic Chinese parachute ankle brace. Annals of translational medicine. 2021, 9(4): 318.

Point 3: 3--Results paragraph need more discussions. The authors show the behavior without discussion.

Response 3: Thanks. It will be clearer if the discussion is added to the description of the Results section. However, Figure V2 showed the requirements of Bioengineering-template. Just like the question of “point 1”.

Figure V2 Bioengineering-template

Point 4: 4-Conclusion must be rewritten as a separated  point. 

Response 4: Thanks. We will rewrite the Conclusion section as a separated point.

Reviewer 2 Report

General comment 

The authors evaluated  the ankle lateral ligament injury mechanism and the protective effect of different ankle braces during the landing phase in paratroopers individuals. The study covers an interesting topic, the rationale is well established and the procedures are described in details. However, there are some points that should be addressed. I hope that my comments will be useful to improve the scientific quality of the manuscript.

Introduction

·         Lines 45: please consider to expand the contribution of postural alignment as another intrinsic factor

Methods 

·         Line 75: Please, provide a table regarding subjects’ characteristics

·         Line 91: please specify if paratroopers have conducted a familiarization procedure related to jumping task

·         Line 149: please provide a paragraph regarding the statistical analysis conducted   

 Discussion 

·         Line 234: Please insert a paragraph stating the main practical applications derived from the Authors’ results. Practitioners and sport scientists may benefit from that. It could be also interesting to discuss the role of muscle tuning and impact forces (Nigg & Wakeling 2001;29(1):37-41; Boyer & Nigg 2006 Dec;128(6):815-22; Wakeling et al. 2001 Sep;91(3):1307-17)

Author Response

Response to Reviewer 2 Comments

Dear Reviewer:

Thanks for the editors and reviewers suggestion. We will make point-by-point responses to your comments. In addition, the revised content will be shown in Red font corresponding to the revised manuscript.

General comment 

The authors evaluated  the ankle lateral ligament injury mechanism and the protective effect of different ankle braces during the landing phase in paratroopers individuals. The study covers an interesting topic, the rationale is well established and the procedures are described in details. However, there are some points that should be addressed. I hope that my comments will be useful to improve the scientific quality of the manuscript.

Response: Thanks for your suggestion. We will carefully revise the whole manuscript.

Introduction

Point 1: Lines 45: please consider to expand the contribution of postural alignment as another intrinsic factor

Response 1: Thanks. We will add the postural alignment as another intrinsic factor. In fact, the postural alignment, muscle strenght, articular stability as the intrinsic factor was very important to maintain landing balance of paratroopers.

Methods 

Point 2: Line 75: Please, provide a table regarding subjects’ characteristics

Response 2: Thanks for your suggestion. A total of 16 adult male paratroopers were from the Chinese Air Force. It was also supported by the program of Chinese Air Force. We add the S.D in Revised manuscript as following “71.3±5.4 Kg and 25±1.8 years old”.

Mean value (±S.E) of Subjects’ characteristics

Sex

Career

Number (Size)

Body Weight (Kg)

Age (year)

Male

Paratrooper

N=16

71.3±5.4

25±1.8

Point 3:   Line 91: please specify if paratroopers have conducted a familiarization procedure related to jumping task

Response 3: Yes. A total of 16 adult male paratroopers were from the Chinese Air Force. It was also supported by the program of Chinese Air Force. Their landing positions are all performing the Half-Squat Parachute Landing in China [1]. Of course, there are the different landing position in United States, Israel and the United Kingdom. We recruited paratroopers who were subjects with real-world combat experience from Chinese Air Force. Some researchers have also compared the effects of landing positions of paratroopers and basketball players.

[1]. Niu, W.X.; Wang, Y.; Yao, J.; Zhang, M.; Fan, Y.B. Consideration of Gender Differences in Ankle Stabilizer Selection for Half-Squat Parachute Landing. Aviation space and environmental medicine, 2011, 82(12), 1118-1124.

Point 4: Line 149: please provide a paragraph regarding the statistical analysis conducted 

Response 4: Thanks. Do you mean to provide the description of statistical analysis in section 2.2? Line 149 showed the process of the finite element model. whether it is continuous with the previous paragraph? Please confirm whether the description of the statistical analysis was added in section 2.2. The description of statistical analysis as following “Quantitative data would firstly be shown by Mean value. Their landing time had a tendency towards consistency due to the of the professional subjects. Each subject performed three jump down from different height platforms. Means and standard deviations were used for the analysis of the dynamic stability.”

Discussion 

Point 5:   Line 234: Please insert a paragraph stating the main practical applications derived from the Authors’ results. Practitioners and sport scientists may benefit from that. It could be also interesting to discuss the role of muscle tuning and impact forces (Nigg & Wakeling 2001;29(1):37-41; Boyer & Nigg 2006 Dec;128(6):815-22; Wakeling et al. 2001 Sep;91(3):1307-17)

Response 5: Thanks for your suggestion. We will discuss the practical applications from the results based on the previous studies. A paragraph was inserted in here as following “Despite ankle braces could reduce the ankle ligament sprains [28], this study showed the the limited protection of ankle braces with the increase of varus incline angle. From the Figure 5 and Figure 6, both strains and stresses of CFL, ATFL and PTFL had a tendency to increase gradually. It was supported the concept that the ATFL and CFL function together at all positions of ankle flexion to provide stability of the lateral ankle [35]. Of course, the muscle tuning was also important to maintain stability of soft tissue (ankle ligaments) [36, 37]. For the stress and strain of PTFL, there was a tendency to decrease, especially in protection of the outer ankle brace [30]. This was related to increase ATFL strain in external rotation of ankle joint [35]. It was also shown that the lower extremity muscle activity is adjusted in reaction to impact forces with the goal of minimizing soft-tissue vibrations [38]. Therefore, paratrooper landing impact would benefit from the ankle brace [30], landing posture [8], the lower extremity muscle tuning [36-38]. ”

[36].Wakeling, J; Nigg, B. Impact Forces and Muscle Tuning: A New Paradigm. Exercise and Sport Sciences Reviews. 2001, 29(1): 37-41.

[37]. Boyer, K.A; Nigg, B.M. Muscle tuning during running: implications of an un-tuned landing. Journal of biomechanical engineering. 2006, 128(6):815-822.

[38]. Boyer, KA; Nigg, BM. Muscle activity in the leg is tuned in response to impact force characteristics. Journal of Biomechanics. 2004, 37(10): 1583-1588.

Reviewer 3 Report

General Comments:

The goal of this investigation was to examine ankle ligament loading when landing from a jump onto an inclined surface.  Specifically, they appear to be interested in paratroopers landing, but it isn’t clear what the rationale was for this.  There is very little detail provided about paratroopers, their ankle injuries, and why an inclined surface is necessary.  They collected some ground reaction force data of paratroopers landing from different heights, all on a flat surface.  No indication of obtaining human subjects approval was provided.  It also isn’t clear what instructions were provided to them to ensure consistent landings from person-to-person.  It appears the only thing the landing data was used for was to obtain time to stability.  It isn’t clear if, or how the ground reaction forces were used in a dynamic FE model.  Only static data was used to validate the model, and this was from a person standing on a 20 degree incline.  Why only the 20 degree incline?  Also, when landing a person does not land flat footed, they first land on their forefoot and eventually stand flat-footed at the end.  How were the dynamic ground reaction forces and dynamic motions of the foot incorporated into the model?  They make reference to loads equal to have bodyweight.  This makes me very concerned about whether dynamic loads were used, or if only static loads.  Also, since they didn’t measure people landing on inclined surfaces, how were the horizontal ground reaction forces estimated.  The F-scan during static standing only provides a pressure distribution normal to the surface of the foot.  It also isn’t clear how the elastic and rigid braces differed from each other within this model.  Overall, too many questions about the FE model to evaluate whether their results are realistic and trustworthy.  While cadaveric data is limiting, it seems like their FE model should have been validated against published cadaveric data before using a novel FE model on a very complex dynamic task.

Specific Comments:

Title

11)      ‘the’ not necessary.

22)      Should be ‘an’ not ‘a’.

Abstract

11)      Line 18, I’m assuming there were no differences in dynamic stability time between any of the conditions.  Would be helpful for you to explicitly state this.

Introduction

12)      Line 31-2, ‘Ankle ligament injuries are also reported annually in US arm paratroopers [3].’  This is a very vague statement.

23)      Line 34, ‘The cost for treatment and rehabilitation per sprain is also higher [5].’  Higher than what?

34)      Line 34-36, you say that sudden ankle inversion when landing on an inclined plane has not been well addressed.  Why does it need to be addressed, is this a common mode of ankle sprain?

45)      Line 49, is pronation and supination different from eversion and inversion?  If not, should be consistent with how ankle motion is described.

56)      Line 62, is a semi-rigid brace different from the hard plastic brace discussed earlier in the paragraph?

67)      Line 68, why is it important that the previous research couldn’t detect and investigate conditions similar to parachute landing?  Is this something you’re interested in?

78)      Line 70-73, you list aims in this paragraph, but no hypotheses.  Would be stronger if you developed a hypotheses driven research study through the structure of your Introduction.

89)      Line 73, still not sure why you are interested in parachute landings on an inclined platform.

Materials and Methods

11)      No mention of human subjects approval.  This must be obtained in order to publish your results.

22)      Line 77-78, please provide standard deviation along with the mean body mass and age of your subjects.

33)      Line 79, were ankle braces worn in all conditions?  I thought one of the conditions did not include use of an ankle brace.

44)      Line 80, what do you mean by ‘jumped’ from different height platforms.  Did they jump up, or just jump forward with minimal additional height gain from the platform?

55)      Line 80, did you filter the ground reaction forces or were they raw data?

66)      When subjects jumped, did they have a boot over the ankle braces?  What type of boot?

77)      What instructions were provided to them about landing?  Were they required to land with hip and knees flexed as depicted in Figure 1?

88)      Line 82-87, not sure how this part of the paragraph fits with the previous.  Did they wear the F-scan system during their jumps?  Did they land on a 20 degree inclined platform?

99)      Line 94-95, results should not be reported in the Methods.

110)   Line 97-100, I’m not sure how to interpret this plot.  Looks like peak forces are higher with greater platform heights.  Not sure where stability time comes into this, since stability is based on the standard deviation.  There are no standard deviations reported in Figure 2.

111)   Line 109-13, do you mean figure 3?

112)   Line 141, how does the elastic brace differ from the outer ankle brace?  This table has a single line for ankle brace.

113)   Line 147, why was a load of half bodyweight used?  This would only be the load when standing, not during dynamic conditions.

114)   How was the dynamic ground reaction force data coupled with the FE model to produce accurate dynamic loading of the model?  The location of the ground reaction force on the foot changes as you land, plus, the force plate provides a singular force in each direction, it does not provide force distribution across the bottom of the foot, which would be needed.

Results

1)      Line 151-158, these are static standing pressures.  You’re evaluating a dynamic task.  What faith do we have that these static results will produce accurate dynamic results?  You are blaming differences on the resolution of the F-scan, maybe you should change the results of your model to report averages over a similar 25 mm2 area.

2)      Why did you only collect F-scan data when on the 20 degree platform.  Would have been helpful to validate in all incline positions.

3)      Do you have any data from your human subjects landing on the 0 degree platform that can be compared to your FE model on a 0 degree incline to help validate the results of your model?

Discussion

1)      Without a better understanding of how the FE model utilized the dynamic conditions of landing on an inclined surface it is hard to comment on the discussion.  Landing is very dynamic, but it appears that it was treated as a highly static movement.  Only very simplistic static measures were used to validate the model, and not clear how the dynamic model was run.  For example, in the limitations, line 257, they indicate that the Achilles tendon was loaded with half of one foot bearing weight.  Does this mean it was loaded with ½ bodyweight, or just that it was assumed a symmetric force distribution?  Forces during landing are much higher than bodyweight.

Author Response

Response to Reviewer 3 Comments

Dear Reviewer:

Thanks for the editors and reviewers suggestion. We will make point-by-point responses to your comments. In addition, the revised content will be shown in Red font corresponding to the revised manuscript.

General Comments:

The goal of this investigation was to examine ankle ligament loading when landing from a jump onto an inclined surface.  Specifically, they appear to be interested in paratroopers landing, but it isn’t clear what the rationale was for this.  There is very little detail provided about paratroopers, their ankle injuries, and why an inclined surface is necessary.  They collected some ground reaction force data of paratroopers landing from different heights, all on a flat surface.  No indication of obtaining human subjects approval was provided.  It also isn’t clear what instructions were provided to them to ensure consistent landings from person-to-person.  It appears the only thing the landing data was used for was to obtain time to stability.  It isn’t clear if, or how the ground reaction forces were used in a dynamic FE model.  Only static data was used to validate the model, and this was from a person standing on a 20 degree incline.  Why only the 20 degree incline?  Also, when landing a person does not land flat footed, they first land on their forefoot and eventually stand flat-footed at the end.  How were the dynamic ground reaction forces and dynamic motions of the foot incorporated into the model?  They make reference to loads equal to have bodyweight.  This makes me very concerned about whether dynamic loads were used, or if only static loads.  Also, since they didn’t measure people landing on inclined surfaces, how were the horizontal ground reaction forces estimated.  The F-scan during static standing only provides a pressure distribution normal to the surface of the foot.  It also isn’t clear how the elastic and rigid braces differed from each other within this model.  Overall, too many questions about the FE model to evaluate whether their results are realistic and trustworthy.  While cadaveric data is limiting, it seems like their FE model should have been validated against published cadaveric data before using a novel FE model on a very complex dynamic task.

Response: Thanks for your comments. This comment includes all of the following questions. To make it clearer, we will make point-by-point responses to each comments.

Specific Comments:

Title

Point 1: 11) ‘the’ not necessary.

22) Should be ‘an’ not ‘a’.

Response 1: Thanks for your suggestion. We will delete the “the”. The use of “a” is too careless. It will be revised for “an”

Abstract

Point 2: Line 18, I’m assuming there were no differences in dynamic stability time between any of the conditions.  Would be helpful for you to explicitly state this.

Response 2: Thanks. The dynamic stability time corresponds to three heights (52cm, 82cm and 152cm). From the Figure 2, the curve trend of stability time was similar. The purpose of dynamic stability time test is is used for the boundary condition of foot finite element. We will revised the expression in Revised manuscript.

Introduction

Point 3: Line 31-32, ‘Ankle ligament injuries are also reported annually in US army paratroopers [3].’  This is a very vague statement.

Response 3: We will refine the statement as following. “Ankle ligament injuries are also reported annually in US arm paratroopers. It account for 30-60% of parachuting related injuries in the military [3].”

Point 4: Line 34, ‘The cost for treatment and rehabilitation per sprain is also higher [5].’  Higher than what?

Response 4: Thanks. We will refine the statement as following. “The cost for treatment and rehabilitation per sprain is also higher, and the dollar costs ranged from $318 to $941 per patient [5].”

Point 5: Line 34-36, you say that sudden ankle inversion when landing on an inclined plane has not been well addressed.  Why does it need to be addressed, is this a common mode of ankle sprain?

Response 5: Thanks. Although the title is “ Effect of different protection on lateral ankle during landing: an instantaneous impact analysis”, the study of manuscript involved ankle injury during paratroopers landing. Parachutist landing may occur at night. The ground is complex conditions, for example, gravel, slope, and so on. This will increase the risk of ankle injury [3]. To decrease the risk of ankle injury, this article evaluates the protective effect of different protective braces. The Introduction section also showed the ankle inversion injury was main risk factor in a prospective study of ankle injury risk factors [4]. During the actual campaign and daily training, the sudden ankle inversion was often happened. Therefore, we assessed the main risk factor (ankle inversion) based on the common mode of ankle sprain.

[3]. Schmidt, M.D.; Sulsky, S.I.; Amoroso, P.J. Effectiveness of an external-the-boot ankle brace in reducing parachuting related ankle injuries. Injury Prevention, 2005, 11(3), 163-168.

[4]. Judith, F.; Denise, M.; Per, A.F.H.; Renström, M.D.; Saul, T.; Bruce, B. A Prospective Study of Ankle Injury Risk Factors. American Journal of Sports Medicine, 1995, 23(5), 564-570.

     Point 6: Line 49, is pronation and supination different from eversion and inversion?  If not, should be consistent with how ankle motion is described.

Response 6: Thanks for your careful review. Yes, it is different. The subtalar joint is complex three-axis motion structure, including eversion/inversion, dorsiflexion/plantarflexion and abduction/adduction (Figure V1). In fact, the movement posture of forefoot and midfoot is transmitted through the subtalar joint. Movement and sliding occur simultaneously on the articular surface of the talus, especially when motion is generated.

Figure V1. Ankle movement

    Point 7: Line 62, is a semi-rigid brace different from the hard plastic brace discussed earlier in the paragraph?

Response 7: Yes, it is different. Figure V2 showed the semi-rigid ankle brace in army. Semi-rigid means to add spring bars (figure V2b) in red area of Figure V2a. The sping has ability of the elastic deformation, vibration absorption and impact resistance. Whereas the hard plastic (Figure V2c) has the poor ability of the plastic deformation. To some extent, the larger deformations can cause secondary injury to the ankle joint.

Figure V2 Semi-rigid brace

 Point 8: Line 68, why is it important that the previous research couldn’t detect and investigate conditions similar to parachute landing?  Is this something you’re interested in?

Response 8: Thanks. Ankle injuries rank first among all paratroopers landing injuries. There have been similar reports in the United States, Israel and the United Kingdom in Introduction section. Of course, ankle injury is the same situation in China parachute landing injurys. The study is also supported by the program of Chinese Air Force. In combat or training, the parachuter can still get injured while wearing ankle guards. Therefore, we are interested in the biomechanism of ankle joint injury and brace structure.

Point 9: Line 70-73, you list aims in this paragraph, but no hypotheses.  Would be stronger if you developed a hypotheses driven research study through the structure of your Introduction.

Response 9: Thank you for your suggestion. We will add a hypotheses through the structure of Introduction section. As following “There are more components to the sprained ankle inversion than risk factor to the other ankle injury. In an effort to further avoidance risks involved in parachute training, the different protection of ankle brace was applied in the initial military parachute descents. The mechanism of injury is unclear, due to multicomponent involvement of the ligamentous injury, sprained ligaments, articular contusion, and complexed landing platform, all of which are most often underrepresented in current literature.”

    Point 10: Line 73, still not sure why you are interested in parachute landings on an inclined platform.

Response 10: Thanks. The study is also supported by the program of Chinese Air Force. In combat or training, the parachuter can still get injured while wearing ankle guards. Therefore, we are interested in the biomechanism of ankle joint injury and brace structure. It is helpful to design the new ankle brace structure of paratroopers

Materials and Methods

Point 11: No mention of human subjects approval.  This must be obtained in order to publish your results.

Response 11: Thanks for your suggestion. We have human subject approval. The  signed the informed consent of the participats had been send to the Email of Bioengineering on 19 Sep, 2022. As following:

Point 12: Line 77-78, please provide standard deviation along with the mean body mass and age of your subjects.

Response 12: Thanks. We will add the standard deviation along with the mean body mass and age of your subjects in Revised manuscript.

Point 13: Line 79, were ankle braces worn in all conditions?  I thought one of the conditions did not include use of an ankle brace.

Response 13: Yes. No ankle brace was also included from the different height platforms. We will add the “and no braces” in Revised manuscript.

Point 14: Line 80, what do you mean by ‘jumped’ from different height platforms.  Did they jump up, or just jump forward with minimal additional height gain from the platform? did you filter the ground reaction forces or were they raw data?

Response 14: Thanks. We may have the ambiguous expressions. The “jumped” did not include “jump up”. It should be revised for “jumped down”. The height of platform was survey and study of out team [8]. On the other hand, safety of participants must be considered in measurement. The raw data was from the standing on  platform to jump down force plate. It was about a few seconds. So we saved the time from the moment the foot hit the ground to full stability. The ground reaction force definitely increases with the height of jump. That is also true. We are concerned with the stability time of landing here. It is very important for ankle stability.

[8]. Niu, W.X.; Wang, Y.; Yao, J.; Zhang, M.; Fan, Y.B. Consideration of Gender Differences in Ankle Stabilizer Selection for Half-Squat Parachute Landing. Aviation space and environmental medicine, 2011, 82(12), 1118-1124.

Point 15:  When subjects jumped, did they have a boot over the ankle braces?  What type of boot? What instructions were provided to them about landing?  Were they required to land with hip and knees flexed as depicted in Figure 1?

Response 15: Thanks. They jumped down with wearing a boot and the different ankle braces. The type of boot was provided by the Chinese Air Force. Why did we choose the position of Figure 1? It is the rule of action in Chinese Air Force. Chinese paratroopers have traditionally performed the half-squat parachute landing, which was different from the wellknown parachute landing fall. The parachute landing fall is widely known in many countries [3]. However, in some countries such as China and Russia, paratroopers perform a standard half-squat parachute landing, which is different from the well-known parachute landing fall [4,5]. Decker et al. found that women chose to land in a more erect posture to maximize the energy absorption from the lower extremity joints most proximal to ground contact [6]. Therefore, we choose the posture of Figure 1. Thanks again for your interest.

[3]. Bricknell MC, Craig SC. Military parachuting injuries: a literature review. Occup Med (Lond) 1999, 49: 17 -26.

[4]. Niu W, Wang Y, He Y, Fan Y, Zhao Q. Biomechanical gender differences of the ankle joint during simulated half-squat parachute landing. Aviat Space Environ Med 2010, 81: 761-767.

[5]. Xie X, An XZ, Tian YJ. A prospective study of 1,795 injuries of parachute landing . Journal of Preventive Medicine of Chinese People’s Liberation Army 2004, 22: 114-115 (Chinese) .

[6]. Decker MJ, Torry MR, Wyland DJ, Sterett WI, Steadman JR. Gender differences in lower extremity kinematics, kinetics and energy absorption during landing. Clin Biomech (Bristol, Avon) 2003, 18: 662-66 9.

Point 16: Line 82-87, not sure how this part of the paragraph fits with the previous.  Did they wear the F-scan system during their jumps?  Did they land on a 20 degree inclined platform?

Response 16: Line 82-87 showed the other two parts work of paragraph. Figrue V3a showed how to obtain the point cloud of the ankle brace and boot. Figure V3b showed the plantar pressure of subjects standing on a 20° inclined platform by F-scan system measurement. I wonder if another paragraph is needed. Please give me some suggestions. Thanks.

Figure V3. (a) point cloud data of ankle brace and boot; (b) plantar pressure of 20° inclined platform by F-scan

Point 17:  Line 94-95, results should not be reported in the Methods.

Response 17: Thanks for your suggestion. We will delete the results in the Method section.

Point 18: Line 97-100, I’m not sure how to interpret this plot.  Looks like peak forces are higher with greater platform heights.  Not sure where stability time comes into this, since stability is based on the standard deviation.  There are no standard deviations reported in Figure 2.

Response 18: Thanks. Figure 2 showed the stability time of landing. The stability time of langing was defined as the time course from the moment of foot touch ground to foot standing stability. The ordinate of Figure 2 showed the ground reaction force tendency with time change. Reaction force was from the relationship between the body weight and jump height. Just think about the moment of foot touch the ground, was the reachtion force maximum? No, it was not. Therefore, the abscissa of Figure 2 shows that the stability time starts from “0”. With increasing contact area and weight impact, the ground reaction force will get a peak. How to determine the steady state? Subjects eventually standed on a pressure plate. From the curve of force-time, the vertical upward force is equal to the body weight. Is that why the final curve has a gap in ordinate of Figure 2.

Point 19: Line 109-13, do you mean figure 3?

Response 19: Yes, it was careless. We will revise figure 2 for figure 3.

Point 20:  Line 141, how does the elastic brace differ from the outer ankle brace?  This table has a single line for ankle brace.

Response 20: Thanks for your suggestion. The elastic brace like a sock was worn on the surface of the foot. The outer ankle brace was worn on the surface of the boot. The material of the elastic brace is a composite material with high elasticity and breathable function. The elastic brace and foot is defined as “Tie” in the contact relationship of the ABAQUS software. The material of the outer ankel brace is a slightly harder, but has some elasticity. The outer ankle brace and foot is defined as “Contact” in the contact relationship of the ABAQUS software.

Point 21: Line 147, why was a load of half body weight used?  This would only be the load when standing, not during dynamic conditions.

Response 21: Thanks. In fact, no matter jumping or playing basketball, doing exercise. Both feet load the body weight at the same time when landing on ground. Each foot bears half of the body weight in Theoretical Mechanics. Which is also used by many researchers [1-10]. It was quite right to consider the dynamic condition of loading. Impact analysis requires mass block and loading time. The half body weitht and the curve of stability time V ground reaction force were used in our FE model. This simulated the dynamic condition during paratrooper landing on ground.

[1]. Junchao Guo#, Lizhen Wang, Zhongjun, Wei Chen, Yubo Fan*. Biomechanical behavior of valgus foot in children with cerebral palsy: A comparative study, Journal of Biomechanics, 2015, 48: 3170-3177.

[2]. Junchao Guo#, Lizhen Wang, Wei Chen, Chengfei Du, Zhongjun Mo, Yubo Fan*. Parametric study of orthopedic insole of valgus foot on partial foot amputation , Computer Methods in Biomechanics and Biomedical Engineering, 2016, 19(8): 894-900.

[3]. Junchao Guo#, Lizhen Wang, Zhongjun Mo, Wei Chen, Yubo Fan*. Biomechanical analysis of suture locations of the distal plantar fascia in partial foot , International Orthopaedics, 2015, 39(12): 2373-2380.

[4]. Rui Mao, Junchao Guo#, Chenyu Luo, Yubo Fan, Jianmin Wen, Lizhen Wang*. Biomechanical study on surgical fixation methods for minimally invasive treatment of hallux valgus. Medical Engineering and Physics, 2017,46: 21-26.

[5]. Junchao Guo#; Xiaoyu Liu; Xili Ding; Lizhen Wang; Yubo Fan*. Biomechanical and mechanical behavior of the plantar fascia in macro and micro structures. Journal of Biomechanics, 2018, 76: 160-166. 

[6]. Junchao Guo#; Lizhen Wang; Rui Mao; Cheng Chang; Jianmin Wen*; Yubo Fan*. Biomechanical evaluation of the first ray in pre-/post-operative hallux valgus: A comparative study. Clinical Biomechanics, 2018, 60:1-8.

[7]. Luo, YX#; Luo, CY; Cai, YH; Jiang, TY; Chen, TH; Xiao, WY; Guo, JC*; Fan, YB*. Analysis of Bone Mineral Density/Content of Paratroopers and Hoopsters. Journal of Healthcare Engineering, 2018 doi:10.1155/2018/6030624.

[8]. Su, HL; Mo, ZJ; Guo, JC et al. The effect of arch height and material hardness of personalized insole on correction and tissues of flatfoot

[9]. Wong, Duo Wai-Chi; Wang, Yan; Niu, Wenxin; Zhang, Ming. Finite element analysis of subtalar joint arthroereisis on adult-acquired flexible flatfoot deformity using customised sinus tarsi implant. J Orthop Translat, 2021, 27: 139-145.

[10]. Peng, Yinghu; Wong, Duo Wai-Chi; Chen, Tony Lin-Wei; Wang, Yan; Zhang, Ming. Influence of Arch Support Heights on the Internal Foot Mechanics of Flatfoot during Walking: A Muscle-Driven Finite Element Analysis. Computers in biology and medicine, 2021, 132: 104355.

Point 22: How was the dynamic ground reaction force data coupled with the FE model to produce accurate dynamic loading of the model?  The location of the ground reaction force on the foot changes as you land, plus, the force plate provides a singular force in each direction, it does not provide force distribution across the bottom of the foot, which would be needed.

Response 22: Thanks. It is a bit like the above question The half body weitht and the curve of stability time and ground reaction force were used in our FE model. This simulated the dynamic condition during paratrooper landing on ground. How to perform? It was realized to use amplitude curves in “Load” or “Interaction” module of ABAQUS software. The curve of stability time and ground reaction force in Figure 2 was applied to the loading of FE model. The force plate provided the force of the vertical direction. The curve of Figure 2 was also adapting to the location of the ground reaction force on the foot changes as you land.

Results

Point 23: Line 151-158, these are static standing pressures.  You’re evaluating a dynamic task.  What faith do we have that these static results will produce accurate dynamic results?  You are blaming differences on the resolution of the F-scan, maybe you should change the results of your model to report averages over a similar 25 mm2 area. Why did you only collect F-scan data when on the 20 degree platform.  Would have been helpful to validate in all incline positions.

Response 23: Thanks for your suggestions. This study focuses on the biomechanical effects of different landing slop on the internal tissues of ankle by a finite element model. Although we were evaluating a dynamic task, the finite element model must be valided by the element amounts and convergence test. After this treatment, the model used for dynamic analysis is valid. Just like a screw-connected mechanism, if the connection is not strong under static conditions, how can dynamic experiments be conducted? The element amounts of mesh quantity is not arbitrary. In fact, we do the the convergence test of the FE model. To test the convergence of the FE model, the displacement of talus measured was used to similar consideration for the other bones. A reference point of talus surface was selected and the top surface of tibia was loaded a uniformly distributed load of 0.3MPa. Five different amounts-146509 (A), 128183 (B), 104219 (C), 91076 (D), and 70843 (E) elements were compared for their corresponding displacements. By setting the displacement of talus to 146509 elements as a reference value, the errors with the total number of elements were within 1.8%. In this model, a total of 91076 elements were selected based on the small relative displacement error of 0.37% for the talus. Take the talus element numbers for an example, the displacement of talus corresponding to five different element amounts was tested in R-Fig.1. The results showed the displacement error of the different elements amounts. The displacement error was B (1.7%), C (2.0%), D (0.37%) and E (1.3%) compared with the A element amounts (Figure V4). If the difference of the results is within the allowable range of error, then it is acceptable. Then the result can also prove the reliability of the data. After all, plantar pressure was obtained by two methods of the FE model prediction and F-scan measurement. There were some limitations. We choose 20 degree data of F-scan measurement and software analysis to validate the FE model. In fact, we can also choose the other degree. No matter it's the FE model of head, cervical spine, hip joint, knee joint. But the end result is to determine the validity of the FE model. However, our FE model was tested by the element amounts and convergence test. The validation of FE model is also currently used by many researchers [1-10]. The validation method what many researchers have used.

Figure V4. Take the talus element numbers for an example to test the convergence of the finite element model. A, B, C, D and E corresponding to five different amounts-146509 (A), 128183 (B), 104219 (C), 91076 (D), and 70843 (E) elements.

Point 24: Do you have any data from your human subjects landing on the 0 degree platform that can be compared to your FE model on a 0 degree incline to help validate the results of your model?

Response 24: Thanks for your suggestion. We will revise from Figure 5 to Figure 7 in Revised manuscript. It is very good question. In fact, 0 degree incline is the horizontal plane. Compared with the 0 degree incline, peak stress and strain of CFL had a large increase from the model prediction.

Discussion

Point 25: Without a better understanding of how the FE model utilized the dynamic conditions of landing on an inclined surface it is hard to comment on the discussion.  Landing is very dynamic, but it appears that it was treated as a highly static movement.  Only very simplistic static measures were used to validate the model, and not clear how the dynamic model was run.  For example, in the limitations, line 257, they indicate that the Achilles tendon was loaded with half of one foot bearing weight.  Does this mean it was loaded with ½ bodyweight, or just that it was assumed a symmetric force distribution?  Forces during landing are much higher than bodyweight.

Response 25: Thanks. Each foot bears half of the body weight in Theoretical Mechanics. Which is also used by many researchers [1-10]. It was quite right to consider the dynamic condition of loading. Impact analysis requires mass block and loading time. The half body weitht and the curve of stability time V ground reaction force were used in our FE model. This simulated the dynamic condition during paratrooper landing on ground. This study focuses on the biomechanical effects of different landing slop on the internal tissues of ankle by a finite element model. This simulated the dynamic condition during paratrooper landing on ground. How to perform? It was realized to use amplitude curves in “Load” or “Interaction” module of ABAQUS software. The curve of stability time and ground reaction force in Figure 2 was applied to the loading of FE model. Therefore, our simulation process is also a dynamic process simulated by curve of time and force profiles (Figure 2). We will add the Discussion section.
